# Learning Counterfactual Explanations for Recommender Systems

## ABSTRACT

We introduce the **L**earning to e**X**plain **R**ecommendations (**LXR**) framework, a post-hoc, model-agnostic framework for counterfactual explanations of recommender systems. LXR can work with any differentiable recommender and learns to score the importance of users' personal data with respect to a recommended item. The framework's objective employs a novel self-supervised counterfactual loss term that seeks to spotlight the user data most instrumental in the recommendation of an item. Additionally, we propose several counterfactual evaluation metrics for assessing explanations in recommender systems. Using these metrics, our results demonstrate LXR's capability to provide counterfactual explanations for various recommendation algorithms across different datasets. LXR's code is publicly available at https://github.com/ExplainingRecommendations/LXR.

## CCS CONCEPTS

• **Computer systems organization** → **Embedded systems**; *Redundancy*; Robotics; • **Networks** → Network reliability.

## KEYWORDS

recommender systems, explanations

**ACM Reference Format:**
Anonymous Author(s). 2023. Learning Counterfactual Explanations for Recommender Systems. In *Proceedings of ACM Conference (Conference'17)*. ACM, New York, NY, USA, 10 pages. https://doi.org/10.1145/nnnnnnn.nnnnnnn

## 1 INTRODUCTION

The growing complexity of artificial intelligence (AI) systems has led to a surge in demand for eXplainable Artificial Intelligence (XAI) methods. As an example, the European Union General Data Protection Regulation (GDPR)[1] dictates that users have a basic "right to an explanation" concerning algorithmic decisions based on their personal information [17]. Similar legislation is under discussion in other countries worldwide. This particularly concerns recommender systems, which utilize user data to generate personalized recommendations. However, despite the importance of explainability in recommender systems, there remains a notable gap in research focusing on model-agnostic counterfactual explanation frameworks for recommender systems.

---

[1]https://www.consilium.europa.eu/en/policies/data-protection/data-protection-regulation/

Counterfactual explanations are explanations that address "what-if" questions, providing insights by comparing the system's recommendations under different conditions. This approach is recognized to be more comprehensible, technically verifiable, and informative to users [52]. In our context, counterfactual explanations entail identifying the elements in a user's personal data that, if altered, would change the recommended item, distinguishing these elements from other parts of the user's data that can be changed without affecting the recommendation.

The naive approach to generating counterfactual explanations involves computationally expensive perturbations in which different subsets of a user's data are modified to observe the resulting changes in the recommendation [60]. This poses a challenge that hinders the ability to efficiently compute counterfactual explanations for multiple items in real-time. Our solution overcomes this obstacle by employing a novel framework in which an *explainer* model learns the changes in a recommender's output with respect to changes in the user's personal data.

We introduce the **L**earning to e**X**plain **R**ecommendations (**LXR**) framework. LXR is a model-agnostic, post-hoc approach for self-supervised learning of counterfactual explanations in recommender systems. LXR's explainer generates counterfactual explanation masks that score the significance of a user's personal data with respect to a given recommendation. Being model-agnostic, LXR can work with any differentiable black-box recommender and does not require knowledge of the recommender's internal architecture or access to its parameters. Its sole prerequisite for the recommender model is its differentiability.

Our contributions can be summarized: (1) We introduce a pioneering, model-agnostic framework specifically designed for efficient computation of counterfactual explanations in recommender systems. A key advantage of LXR, especially when compared to alternative counterfactual explanation techniques, lies in its ability to rapidly generate explanations, bypassing the necessity for intensive computations or perturbations. This proficiency positions LXR as a leading choice for delivering on-demand explanations in real-world systems. (2) LXR's self-supervised optimization process is guided by a novel counterfactual loss term, intricately incorporating any 'black-box' recommender in a distinctive configuration. (3) We put forward a methodology for the counterfactual evaluation of explanatory algorithms in recommender systems. This methodology employs a fresh set of metrics, drawing inspiration from saliency map evaluations in computer vision [9, 10, 37]. (4) Using these metrics, we benchmark LXR against prevalent explanatory approaches, highlighting LXR's superior capacity to explain a variety of recommendation algorithms across multiple datasets.

## 2 RELATED WORK

In recent years, the need for XAI methods has gained significant interest from both the research community and the industry. In the context of recommender systems, it has been shown that transparency and interpretability foster trust in users [8, 22, 45]. Consequently, numerous works have focused on deriving explanations for

recommender systems. While it is not feasible to cover every work, we attempt to provide a concise overview of the most prominent approaches and refer the interested reader to [50, 58] for more on the subject.

We categorize explanation methods for recommender systems into several primary groups. Although this categorization is not mutually exclusive, we find it useful for the sake of our discussion:

**Model-specific methods:** These methods are tailored for specific recommendation algorithms and rely on the unique properties and structures of the explained model. For example, Abdollahi and Nasraoui [1, 2] presented explanation approaches for matrix factorization models (MF) [28]. In this category, we also include recommendation models designed to inherently induce interpretability and reasoning with respect to the model's predictions [7, 16, 33]. For example, ProtoMF [33] is a recommendation algorithm that utilizes user prototypes to enable explainability. LXR inherently differs from the above works since it is a *model-agnostic* framework, designed to work with *any* differentiable recommender algorithm and not tailored to one specific recommendation model.

**Post-hoc methods:** Post-hoc methods are applied *after* a model has been trained in order to generate explanations for its predictions [13, 35, 36, 51]. For example, LIME-RS [35] is a model-agnostic adaptation of LIME [41] which builds a simple surrogate model to approximate the original recommender in order to explain it. FIA (Fast Influence Analysis) [13] is a post-hoc approach for explanations in latent factor models using influence functions [27]. Finally, Tran et al. [51] presented ACCENT (Action-based Counterfactual Explanations for Neural Recommenders for Tangibility), a generalization of [13] for model-agnostic counterfactual explanations in recommender systems. Similar to the above works, the LXR model in this paper belongs to the model-agnostic, post-hoc category.

**Aspect-based methods:** An extensive line of research deals with methods that leverage external item features or 'aspects' [11, 12, 20, 24, 30, 46, 47, 53–55, 57, 59]. For example, [59] proposed the Explicit Factor Model (EFM) that aligns latent factors with explicit features such as color and price to generate explanation sentences. The LXR model differs from this line of work as it does not depend on any item features or aspects to generate explanations.

**Counterfactual methods:** Counterfactual methods [52] operate by addressing "what-if" questions that attempt to assess how the model's recommendations would vary with respect to different changes in the user's personal data. These methods were shown to be more user-friendly and easier to understand [52]. As a consequence, several studies proposed counterfactual explanation techniques for recommender systems [25, 47, 60]. However, these methods usually depend on resource-intensive searches or perturbations leading to increased computational costs that prohibit real-time deployment in a commercial setting. For example, the SHAP method [32], stemming from Shapley values [19] in game theory, computes the marginal contribution of each data element across different perturbations. However, computing perturbations for every explanation hinders employing SHAP in real-world settings where explanations are computed on demand in run-time. In [60], SHAP was applied to recommender systems to create counterfactual explanations using a limited set of 12 explainable features. To overcome the computational challenge, the authors proposed to heuristically select a subset of features on which to perform the

perturbations. In contrast to the above works, the LXR method does not perform perturbations to compute explanations. Instead, LXR overcomes the computational challenge by training a model that employs a novel counterfactual loss function in order to learn the changes in the recommender's output with respect to changes in the user's personal data. At run-time, instead of performing expensive perturbations, LXR generates its explanations via a simple feed-forward inference step.

## 3 THE LXR FRAMEWORK

### 3.1 Problem Setup and Preliminaries

Let $\mathcal{U}$ and $\mathcal{V}$ be the number of users and items, respectively. For a specific user $u$, we denote by $\mathbf{x}_u$ her personal data vector. In this work, we chose to demonstrate LXR to explain implicit feedback Collaborative Filtering (CF) recommenders [23, 38]. Accordingly, $\mathbf{x}_u \in \{0, 1\}^{\mathcal{V}}$ is a binary vector that indicates the historical items consumed by $u$. We note that LXR is a general framework, not limited to CF recommenders. Hence, in general, $\mathbf{x}$ may include any user data.

Let $f_\theta : \{0, 1\}^{\mathcal{V}} \to [0, 1]^{\mathcal{V}}$ be a CF recommender, parameterized by $\theta$, that receives user data $\mathbf{x}$, and outputs a probability distribution over the items. This recommender is already trained and our goal is to explain its predictions with respect to the user data. Hence, we treat the recommender $f_\theta$ as a "black box" that receives user data vector $\mathbf{x}$ as input and produces user-item affinity scores which can be used to rank items and produce personalized recommendations.

An *explainer* $e_\phi : \{0, 1\}^{\mathcal{V}} \times \{0, 1\}^{\mathcal{V}} \to [0, 1]^{\mathcal{V}}$ is a function, parameterized by $\phi$, that receives a user data vector $\mathbf{x}$, and a one-hot vector $\mathbf{y} \in \{0, 1\}^{\mathcal{V}}$, representing the recommended item (target item) to be explained. The explainer's output is an *explanation mask*, a vector $\mathbf{m} \in [0, 1]^{\mathcal{V}}$ that attributes the relevance of each item in the user's history $\mathbf{x}$ with respect to the recommendation of the target item $\mathbf{y}$. Namely, $\mathbf{m} = e_\phi(\mathbf{x}, \mathbf{y})$, and $\mathbf{m}[i]$ represents the importance of the data in $\mathbf{x}[i]$ to the recommendation of $\mathbf{y}$. Note that $\mathbf{y}$, the target item to be explained, is determined according to our choice. The property of an explanation algorithm to produce different explanations for different predictions is known as "discriminability" [60].

Finally, we denote by $\mathbf{x}^m$ the Hadamard product between the user data vector $\mathbf{x}$, and the explainer's mask $\mathbf{m}$. Formally, $\mathbf{x}^m = \mathbf{x} \circ \mathbf{m}$. Based on the explanation scores in $\mathbf{m}$, the vector $\mathbf{x}^m$ is essentially a masked version of the user data $\mathbf{x}$ in which the *most important* elements are kept, while the *least important* elements are removed. Similarly, we denote by $\mathbf{x}^{1-m}$ the Hadamard product between the user data vector $\mathbf{x}$, and the inversed mask $(1 - \mathbf{m})$. Formally, $\mathbf{x}^{1-m} = \mathbf{x} \circ (1 - \mathbf{m})$. Accordingly, $\mathbf{x}^{1-m}$ is a masked version of the user's data $\mathbf{x}$ in which the *most important* elements are removed while the *least important* elements are kept.

### 3.2 LXR Objective

In order to optimize the explainer $e_\phi$, LXR employs a novel self-supervised counterfactual objective as follows:

$$\mathcal{L}_{LXR}(\mathbf{x}, \mathbf{y}) = \mathcal{L}_{pred}(f_\theta(\mathbf{x}^m), \mathbf{y}) + \lambda_{inv}\mathcal{L}_{inv}(f_\theta(\mathbf{x}^{1-m}), \mathbf{y}) + \lambda_{mask}\mathcal{L}_{mask}(m). \quad (1)$$

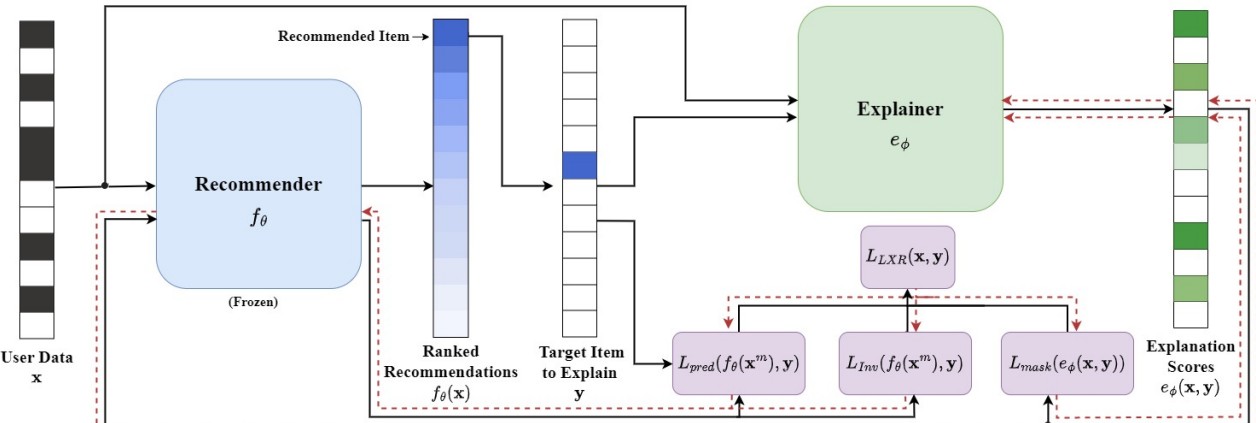

**Figure 1: The LXR framework: The explainer $e_\phi$ is trained to explain the predictions generated by the (frozen) recommender $f_\theta$. Note the backpropagation of the gradients, marked by the red dashed lines traversing through the recommender (yet without updating it), back to the explainer. This unique setup enables the learning of counterfactual explanations for $f_\theta$ in a self-supervised manner while avoiding the need to perform expensive perturbations that prohibit the implementation of other counterfactual explanation techniques in real-world recommender systems.**

The above objective consists of three terms: The first term, $\mathcal{L}_{pred}$, is designed to encourage the explainer to output an explanation mask that leads the recommender to keep the target item $\mathbf{y}$ at the top of the recommendation list, even when it considers the masked version of the user data $\mathbf{x}^m$. The second term, $\mathcal{L}_{inv}$, is designed to suppress the recommendation of $\mathbf{y}$ when the user's data is masked by the inverted mask $(1 - \mathbf{m})$ in which the most important elements are removed. Finally, the goal of the third objective, $\mathcal{L}_{mask}$, is to enforce sparsity on the mask.

According to LXR's design goals above, we experimented with different implementations for the three loss terms in Eq. 1. In this work, we set $\mathcal{L}_{pred}$ to the categorical cross-entropy between $f_\theta(\mathbf{x}^m)$, and the target item $\mathbf{y}$. Hence, $\mathcal{L}_{pred}$ strives to maintain the original recommendations of $\mathbf{y}$ for the masked version of the user input $\mathbf{x}^m$, in which only the most important elements are kept. $\mathcal{L}_{inv}$ was set to $-log(1 - f_\theta(x^{1-m})[y])$ with $y$ representing the index of the target item. Consequently, $\mathcal{L}_{inv}$ strives to minimize the probability score assigned to the target item when the recommender sees $\mathbf{x}^{1-m}$, in which the most important data elements are masked. Finally, $\mathcal{L}_{mask}$ was set to the L1 loss.

We wish to highlight the novelty in LXR's objective from Eq. 1. In essence, $\mathcal{L}_{pred}$ and $\mathcal{L}_{inv}$ are counterfactual loss terms that strive to answer "what-if" questions with respect to potential inputs $\mathbf{x}^m$ and $\mathbf{x}^{1-m}$ induced by the explainer $e_\phi$. To this end, LXR employs an untraditional setup in which the recommender itself, $f_\theta$, is embedded inside the learning objective and takes an active part in the optimization process. Nevertheless, the recommender's weights, $\theta$, are kept frozen, as we do not wish to change the object of our explanations. This unique setup enables the learning of counterfactual explanations for $f_\theta$ in a self-supervised manner. It also avoids the need to perform expensive perturbations that prohibit the implementation of other counterfactual explanation techniques in real-world recommender systems.

### 3.3 LXR Optimization

During LXR's training, for each train user $u$, we set the target item to be explained $v$, as the item with the highest score according to the recommender (excluding the historical items that appear in $\mathbf{x}_u$). We then train $e_\phi$ according to:

$$\phi^* = \underset{\phi}{\operatorname{argmin}} \ \frac{1}{\mathcal{U}} \sum_{u=1}^{\mathcal{U}} \mathcal{L}_{LXR}(\mathbf{x}_u, \mathbf{y}_{t(u)}), \qquad (2)$$

where $\mathcal{L}_{LXR}$ is LXR's objective from Eq. 1, and

$$t(u) = \underset{i \in \{j | \mathbf{x}_u[j]=0\}}{\operatorname{argmax}} f_\theta(\mathbf{x}_u)[i], \qquad (3)$$

is the recommended item to the user.

The LXR optimization process is carried out using stochastic gradient descent. Equipped with the trained explainer $e_{\phi^*}$ (optimized for recommender $f_\theta$), the explanation mask for the pair $(\mathbf{x}, \mathbf{y})$ is given by $e_{\phi^*}(\mathbf{x}, \mathbf{y})$ which requires a simple feed-forward operation and avoids the need to perform perturbations. Figure 1 depicts a schematic illustration of the LXR framework and its optimization.

### 3.4 The Explainer Architecture

The LXR framework, as described above, is versatile and does not rely on either the explainer's or the recommender's architectures. Significantly, the lack of dependence on the recommender's architecture distinguishes LXR as a model-agnostic approach, setting it apart from many current explanation frameworks in recommender systems.

Regarding the explainer's architecture, our experiments indicate that LXR works well using different architectural choices. In this work, the explainer $e_\phi$ was implemented as a neural network that receives a user data vector $\mathbf{x}$ and the target item $\mathbf{y}$ as inputs and applies linear mappings to produce the embeddings $\mathbf{q}_x = \mathbf{W}_0\mathbf{x}$ and $\mathbf{q}_y = \mathbf{W}_1\mathbf{y}$. These embeddings are concatenated to form a

super-vector $\mathbf{q} = [\mathbf{q}_x, \mathbf{q}_y]$ which is fed to an MLP network $g_{\phi_{MLP}}$, with $n$ Tanh activated hidden layers, followed by a sigmoid activated layer with an output size $\mathcal{V}$. The final explanation mask is given by $e_\phi(\mathbf{x}, \mathbf{y}) = g_{\phi_{MLP}}(\mathbf{q}) \circ \mathbf{x}$. The explainer's parameters, $\phi = \{\mathbf{W_0}, \mathbf{W_1}, \phi_{MLP}\}$, were optimized according to the objective in Eq. 2. The final implementation, including hyperparameter settings are further detailed in Sec. 5 and provided in our GitHub repository.

## 4 COUNTERFACTUAL EVALUATION

Quantitative evaluation of explanation algorithms for recommender systems is a challenging task. Many previous models focused on generating aspect-based explanations, with evaluations often performed on datasets containing item reviews from which the items' aspects were extracted [14, 20, 24]. Similarly, Xian et al. [57] evaluated their explanations based on the ability to identify the essential attributes of recommended items. However, these approaches are limited to aspect-based explanation methods where explanations rely on item attributes. In contrast, LXR is a general explanation method that does not necessitate any item information. In fact, in this paper, we demonstrate LXR for CF recommenders, in which the only available information is the user-item interaction matrix.

In the context of post-hoc, model-agnostic explanations for recommender systems, evaluations often focus on measuring model fidelity, as in [20, 35]. However, model fidelity in these works is a coverage metric: $Model\ Fidelity = \frac{|Explainable\ Items \cap Recommended\ Items|}{|Recommended\ Items|}$, which does not consider the actual quality of the explanations. In LXR, we can generate explanations for any recommendation, resulting in a Model Fidelity of 1. Hence, the objective of our evaluations is not about LXR's coverage, but rather its ability to produce accurate counterfactual explanations with respect to the recommended item. Other useful methods for evaluating explanations in recommenders include online experiments [57], user studies [34], and various heuristics related to desirable explanation traits, such as transparency, scrutability, trust, persuasiveness [48, 49].

### 4.1 Perturbation Tests

In this work, we diverge from the aforementioned evaluation methods and present a framework for counterfactual evaluation of explanation algorithms for recommender systems based on a new set of metrics, inspired by saliency map evaluations in computer vision. Given the scarcity of counterfactual evaluation metrics for recommender system explanations, we hope that this aspect of our work would serve as an additional contribution to the community.

Our evaluation metrics draw inspiration from metrics such as the area over the perturbation curve (AOPC) introduced by Samek et al. [42], as well as the Prediction Gap on Unimportant (PGU) and Prediction Gap on Important (PGI) metrics proposed by Agarwal et al. [4] which are commonly employed for reporting explanation results of saliency maps in computer vision [9, 10, 37]. In the case of images, these tests involve gradually removing (e.g., blackening) pixels based on their "explainability" score, either in ascending or descending order while observing the change in the model's prediction. In a *positive* perturbation test the pixels are eliminated in decreasing order of "explainability", hence it is expected that the model's output will change rapidly and significantly. Conversely, in a *negative* perturbation test, pixels are removed in increasing

order of "explainability", hence it is anticipated that the model's output will change slowly and incrementally.

In the context of recommender systems, rather than image pixels, we deal with users' personal data. Specifically, in this work, we focus on CF models in which the user data consists of $\mathbf{x} \in \{0, 1\}^{\mathcal{V}}$, a binary vector that indicates the historical items consumed by the user. User recommendations are organized as a ranked list of items based on the recommender's predictions, where $f_\theta(\mathbf{x}_u)[i]$ is the recommender's affinity score for a user data vector $\mathbf{x}_u$ and an item $i$. Consequently, perturbations involve removing items from the user's vector $\mathbf{x}$ according to their "explainability" score provided by the explainer, while monitoring the recommended item to be explained. Accordingly, in a *positive* perturbation test, the user data is deleted in *descending* order of importance or "explainability" with the expectation that the explained item's score would *decrease* quickly and the explained item would move down the recommendation list. In a *negative* perturbation test, the user data is deleted in *ascending* order, with the expectation that the recommended item being explained would maintain its high user-item score and continue to rank high in the recommendation list.

Building on these insights, we propose four new counterfactual evaluation metrics for explanations in recommender systems. Our metrics employ stepwise perturbations where on each step an additional $\frac{1}{M}$ of the user's data is deleted according to its relevance score obtained from the explainer. $M$ is a positive integer that serves as a granularity factor for the number of perturbation steps. For example, in this work, we set $M = 10$.

Let $\mathbf{x}_u$ be the user $u$'s historical items vector. In a *positive* perturbation test, we define $\mathbf{x}_u^{pos}(m)$ to be the user's $u$ data after removing $\frac{m}{M}$ of her *most important* personal data according to the explainer. Similarly, for *negative* perturbations, we define $\mathbf{x}_u^{neg}(m)$ to be the user's $u$ data after removing $\frac{m}{M}$ of her *least important* personal data according to the explainer. Accordingly, in either a *positive* or a *negative* perturbation test, we perform $m = 1, ..., M$ steps, gradually deleting the user's historical items in *decreasing* or *increasing* order of importance, respectively. At step $m = 0$, the user data is complete i.e., $\mathbf{x}_u^{pos}(0) = \mathbf{x}_u^{neg}(0) = \mathbf{x}_u$. At the last step, $m = M$, the entire user data is deleted i.e., $\mathbf{x}_u^{pos}(M) = \mathbf{x}_u^{neg}(M) = \mathbf{0}$ (the zero vector). For example, in a positive (negative) perturbation test, with $M = 10$, if the number of items in the user history is 50, then at step $m = 7$, we delete the $\lfloor 50 \times \frac{7}{10} \rfloor = 35$ most (least) important items from her user history. Finally, we denote by $rank(\mathbf{x}_u)$ the rank of the explained item according to the recommender system for user $u$ with personal data vector $\mathbf{x}_u$.

Given the notation above, we assess the performance of explanation models by measuring the area under the curve (AUC) with respect to the following counterfactual perturbation tests:

**POS-P@K:** We define a *positive* perturbation test that monitors if the explained item remains in the top $K$ recommendations when user data is deleted in *decreasing* order of importance. Formally, at step $1 \leq m \leq M$, we define $POS\text{-}P@K(m) = \mathbb{1}\left[rank(\mathbf{x}_u^{pos}(m)) \leq K\right]$, where $\mathbb{1}[\cdot]$ is the indicator function. Namely, $POS\text{-}P@K(m)$ is an indicator function that denotes whether the explained item remains at the top $K$ recommendations after deleting $\frac{m}{M}$ of the user's *most important* data. The AUC is given by $AUC\ POS\text{-}P@K =$

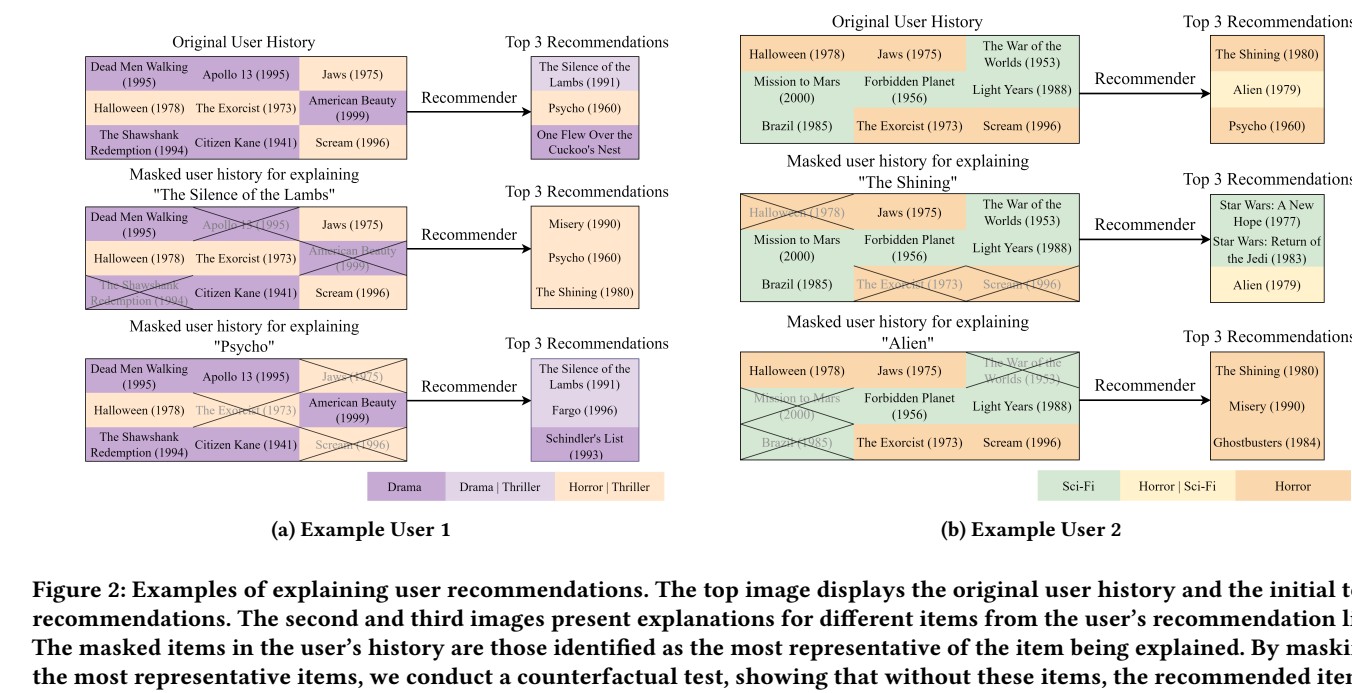

(a) Example User 1

(b) Example User 2

**Figure 2: Examples of explaining user recommendations. The top image displays the original user history and the initial top recommendations. The second and third images present explanations for different items from the user's recommendation list. The masked items in the user's history are those identified as the most representative of the item being explained. By masking the most representative items, we conduct a counterfactual test, showing that without these items, the recommended items would no longer be recommended.**

$\frac{1}{M} \sum_{m=1}^{M} POS@K(m)$. For this metric *lower* values are considered better, as the explained item is expected to drop quickly when the most important user data is deleted first.

**NEG-P@K:** We define a *negative* perturbation test that monitors if the explained item remains in the top $K$ recommendations when user data is deleted in *decreasing* order of importance. Formally, at step $1 \leq m \leq M$, we define $NEG\text{-}P@K(m) = \mathbb{1}\left[rank(\mathbf{x}_u^{neg}(m)) \leq K\right]$, where $\mathbb{1}[\cdot]$ is the indicator function. Hence, $NEG\text{-}P@K(m)$ is an indicator function that denotes whether the explained item remains at the top $K$ recommendations after deleting $\frac{m}{M}$ of the user's *least important* data. The AUC is given by $AUC\ NEG\text{-}P@K = \frac{1}{M} \sum_{m=1}^{M} NEG\text{-}P@K(m)$. For this metric, *higher* values are better, as the explained item is expected to drop slowly when the least important user data is deleted first.

Note that *POS-P@K* and *NEG-P@K* are symmetric versions of each other for the positive and the negative perturbation tests, respectively. However, as we shall see next, these metrics focus on different aspects of the explanations and their results do not necessarily correlate.

**NDCG-P:** Normalized Discounted Cumulative Gain (NDCG) [56] is a common metric in recommender systems evaluation. Hence, we propose a positive perturbation test based on NDCG. At step $1 \leq m \leq M$, we define: $NDCG - P(m) = \frac{1}{\log_2\left(1 + rank(\mathbf{x}_u^{pos}(m))\right)}$, and the AUC is given by $AUC\ NDCG\text{-}P = \frac{1}{M} \sum_{m=1}^{M} NDCG\text{-}P(\mathbf{x}_u^{pos}(m))$. Note that for this metric, *lower* values are considered better, as the expectation is that the explained item will drop quickly when the *most relevant* user data is deleted first.

**DEL-P:** The aforementioned metrics focus on the explained item's rank relative to other items. In contrast, the *deletion* perturbation metric measures the decrease in the recommender's confidence as the user's data is removed in *decreasing* order of importance. Formally, at step $1 \leq m \leq M$, we track the value of $f(\mathbf{x}_u^{pos}(m))[i]$ i.e., the score that the recommender assigns to a user with the vector $\mathbf{x}_u^{pos}(m)$ and the item $i$. The AUC is given by $AUC\ DEL\text{-}P = \frac{1}{M} \sum_{m=1}^{M} f(\mathbf{x}_u^{pos}(m))[i]$. Since the user data is removed in *decreasing* order of importance, in this metric *lower* values are considered better.

**INS-P:** The *insertion* perturbation metric measures the increase in the recommender's confidence as the user's data is *added* in *decreasing* order of importance. Starting from an *empty* user vector, we gradually add items in decreasing order of importance. Note that in terms of AUC, adding the items in *decreasing* order is equivalent to removing the items in *increasing* order of importance, hence INS-P can be formulated as a *negative* perturbation test. Formally, at step $1 \leq m \leq M$, we track the value of $f(\mathbf{x}_u^{neg}(M - m))[i]$ namely, the score that the recommender assigns to the item $i$ and a user with user data $\mathbf{x}_u^{neg}(M - m)$ i.e., an empty user vector to which $\frac{m}{M}$ of $u$'s *most important* items were *added*. The AUC is given by $AUC\ INS\text{-}P = \frac{1}{M} \sum_{m=1}^{M} f(\mathbf{x}_u^{neg}(M - m))[i]$. Consequently, for *INS-P*, *higher* values are considered better.

The metrics above are defined for a single user. We report our evaluations based on the average value over a hidden test set of users as explained next.

# 5 EXPERIMENTAL SETUP AND RESULTS

Our evaluations are based on the MovieLens 1M (ML1M) dataset [18] consisting of 1 million ratings from 6,040 users to 3,883 movies. In addition, in the Appendix we report additional evaluations using the Yahoo! Music dataset [15] and the Pinterest dataset [21].

We focus on the less explored, yet highly relevant, task of explaining CF models based on implicit user-item interactions [23, 38]. Hence, the rating value was disregarded, and the recommenders learned to predict the binary user-item interaction matrix instead. Each dataset was divided into distinct training and testing subsets using user identifiers, following an 80% for training and 20% for testing division. Additionally, we excluded 10% of the users from the training subset for hyperparameter optimization. The evaluations are based on the users in the testing subset. For every user, we utilized her data to produce a ranked list of recommendations and subsequently evaluated the ability to explain her top recommendation.

We consider two CF recommendation models:

**Matrix Factorization (MF):** Matrix factorization models are a class of collaborative filtering techniques that decompose a user-item interaction matrix into lower-dimensional latent factor representations [28]. While these methods have been around for a while, recent reproducibility studies [39, 40] found that MF models, like iALS [23], remain highly competitive compared to many more recent models. We implemented a version of MF in which the model receives a binary encoding of the user's historical items from which it computes the user's latent representation via a simple projection matrix. The user's vector is then multiplied by the item's vector using the inner product followed by a sigmoid in order to produce binary probabilities. The latent dimensionality of the recommender was set to $d = 50$.

**Variational Autoencoder (VAE):** Variational autoencoders (VAEs) are generative models that learn to encode and decode data while simultaneously learning a probabilistic latent representation through optimization of a lower bound on the data likelihood [6]. VAEs have been successfully employed for CF recommendations [31, 43] and have recently been shown to perform extremely well against other baselines in an objective reproducibility study [39]. Hence, we included a recommender model based on a VAE with a similar architecture to that of [31].

The exact implementations of the above recommendation models can be found in our GitHub repository. All the experimentations in this project were done using an Nvidia DGX V100 machine with 4 GPUs.

## 5.1 Baselines

As discussed in Section 2, while much work has been done on explaining recommender systems, very little has been achieved in the way of model-agnostic counterfactual methods. Consequently, we compare LXR against the following baselines:

**Jaccard Similarity (Jaccard):** This baseline produces explanations for a recommended item based on similarity to the items in the user's history using the Jaccard pairwise similarity [5] computed on the users who interacted with both items.

**Cosine Similarity (Cosine):** This baseline produces explanations for a recommended item based on similarity to the items in the user's history using the Cosine pairwise similarity [44] computed on the users who interacted with both items.

**LIME:** LIME approximates the recommender using a locally interpretable, linear surrogate model [41]. In [35], the LIME-RS algorithm was presented as an adaptation of LIME for recommender systems. We employed LIME-RS on the user's historical items. The number of examples was carefully tuned using the validation set.

**SHAP:** SHAP is a counterfactual method based on computationally expensive perturbations [32]. In the context of recommender systems, a naive approach entails experimenting with multiple permutations of the user's historical items. However, the number of possible perturbations is exponential in the number of items, making it impractical for real-world settings where explanations are computed ad-hoc. In [60], SHAP was applied to recommender systems using a limited set of 12 explainable features. In our case, we wish to employ SHAP on the user's historical items (not features). Hence, we grouped each user's items into $K = 20$ clusters using Jaccard similarity and the k-means algorithm. Shapley values were computed for the aggregated item clusters in order to generate explanations. Note that even with just $K = 20$ clusters, this evaluation required intensive computing efforts rendering it impractical for real-world settings.

**ACCENT:** ACCENT (Action-based Counterfactual Explanations for Neural Recommenders for Tangibility) [51] is a state-of-the-art counterfactual explanation framework based on influence functions [27]. It is a model-agnostic generalization of FIA [13] which was originally presented for latent factor models only.

**LXR:** Our proposed model. LXR's linear mappings $\mathbf{W}_0, \mathbf{W}_1 \in \mathbb{R}^{d \times \mathcal{V}}$, were implemented with $d = 20$. The MLP in the explainer includes $n = 2$ hidden layers (the output dimension of the first and second MLP layers was set to $2d$ and $3d$, respectively). The explainer was trained on the training users only, using the Adam [26] optimizer with a learning rate 0.001 and batch size of 64, while monitoring the POS-P@10 metric till convergence (training converged after $\sim 10$ epochs). $\lambda_{inv}$ and $\lambda_{mask}$ were set per dataset, according to a separate validation set. Finally, we set the explainer to the model checkpoint that obtained the best POS-P@10 value across all epochs. We note that it is possible to monitor different metrics, and potentially obtain improved results for each of the explanation metrics from Section 4.

Finally, we also include two ablated versions of LXR as follows:

**LXR$_{pos}$:** An ablated version of LXR in which the $\mathcal{L}_{inv}$ loss term from Eq. 1 was removed. Hence, in this ablated version of LXR the optimization is based on $\mathcal{L}_{pred}$, which strives to maintain the recommended item for the masked input $\mathbf{x}^m$, and $\mathcal{L}_{mask}$, which encourages a sparse mask.

**LXR$_{neg}$:** An ablated version of LXR in which the $\mathcal{L}_{pred}$ loss term from Eq. 1 was removed. Hence, in this ablated version of LXR the optimization is based on $\mathcal{L}_{inv}$, which strives to suppress the recommended item for the inverted masked input $\mathbf{x}^{1-m}$, and $\mathcal{L}_{mask}$, which encourages a sparse mask.

## 5.2 Qualitative Examples

We commence the evaluation section of this paper with a few qualitative examples of counterfactual explanations. Figure 2 presents two examples of recommendations to different users based on the

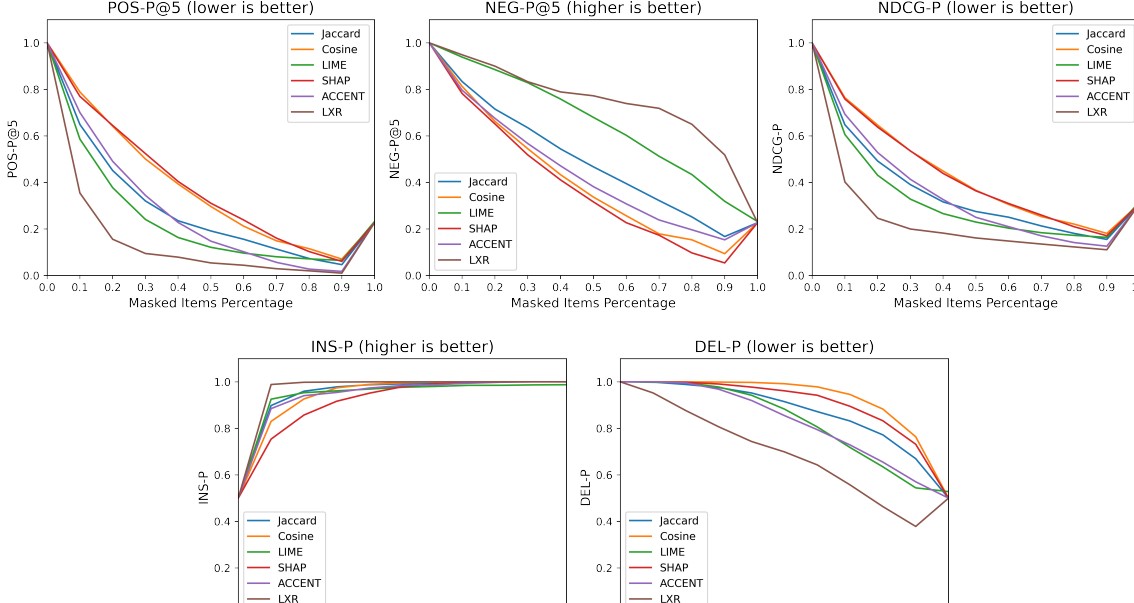

**Figure 3: Perturbation evaluation tests. The figures present *POS-P@5, NEG-P@5, NDCG-P, INS-P* and *DEL-P* perturbation vs. the percentage of masked items in the user history.**

VAE recommender and the ML1M dataset. In each example, the top row displays the user's historical items and the resulting recommendations. The second and third rows for each user display explanations for different items from the user's recommendation list. The masked items in the user's history are those identified as the most responsible for the item being explained. By masking the explaining items, we conduct a counterfactual test, akin to a single *positive* perturbation that shows that without these items, the original recommended item would no longer be recommended. These examples also demonstrate LXR's ability to provide different explanations for different recommended items showcasing LXR's discriminability property [29].

## 5.3 Quantitative Results

We begin with a visual evaluation that presents the perturbation process for different metrics. Figure 3 displays *POS-P@5, NEG-P@5, NDCG-P, INS-P,* and *DEL-P* for LXR and the baselines when explaining the MF recommender on the ML1M dataset. The x-axis depicts the perturbation steps or the percentage of items that are masked. The $m$'th marker on the x-axis signifies the point where $\frac{m}{M}$ of items from the user's history are masked based on their significance score with respect to the item being explained. Keep in mind that for *POS-P@K, NDCG-P,* and *DEL-P lower* values are better, while for *NEG-P@K* and *INS-P@K higher* values are better.

The results in Fig. 3 show that LXR significantly outperforms the baselines across the vast majority of the different perturbation tests. On *POS-P@5, NEG-P@5,* and *NDCG-P* it is clear to see that LIME [35] comes second. In fact, for the *NEG-P@5* test, LXR and LIME seem very close at the first three perturbations until LXR picks up on

the fourth perturbation. Additionally, we note that in the graph for *INS-P*, LXR's lead with respect to the baselines is less dominant. We attribute these findings to the fact that during the LXR training process, we monitored the *POS-P@10* metric, a positive perturbation test. This may have caused LXR to be somewhat biased in favor of positive perturbation tests over negative perturbation tests. We plan to address this in future work by adding negative perturbation tests to our monitoring.

We now turn to a more comprehensive quantitative evaluation using the AUC of the metrics as outlined in Sec. 4. Tables 1-2 display the AUC values for *POS-P@K* and *NEG-P@K* with $K = 5, 10, 20$. Additionally, values for the AUC of *INS-P, DEL-P,* and *NDCG-P* are provided for LXR and baselines in explaining the MF recommender (Tab. 1) and the VAE recommender (Tab. 2). On each evaluation, the best result is emphasized with **bold** fonts, while the second-best result is underlined. The distinction in results between the top model and the next best was statistically verified using a t-test, yielding a p-value of less than 0.05.

For the MF recommender (Tab. 1), LXR variants consistently outperform across all metrics, with LIME [35] as the subsequent contender. In explaining the VAE recommender (Tab. 2), LXR variants lead in most metrics except for *NEG-P@10* and *NEG-P@20*. As explained above, this may be attributed to the fact that during the LXR training process, we monitored the *POS-P@10* metric, a positive perturbation test, which may have caused LXR to be biased in favor of positive perturbation tests over negative perturbation tests. Nevertheless, these results underscore LXR's prowess as a universally applicable counterfactual explanation technique.

**Table 1: AUC values for explaining an MF recommender.**

| Method | k=5 | | k=10 | | k=20 | | INS | DEL | NDCG |
|---|---|---|---|---|---|---|---|---|---|
| | POS | NEG | POS | NEG | POS | NEG | | | |
| Jaccard | 0.25 | 0.46 | 0.27 | 0.48 | 0.32 | 0.54 | 0.98 | 0.85 | 0.31 |
| Cosine | 0.34 | 0.37 | 0.38 | 0.41 | 0.44 | 0.48 | 0.97 | 0.91 | 0.39 |
| LIME | 0.20 | 0.62 | 0.04 | 0.68 | 0.28 | 0.74 | 0.97 | 0.80 | 0.26 |
| SHAP | 0.36 | 0.33 | 0.41 | 0.39 | 0.47 | 0.44 | 0.93 | 0.88 | 0.41 |
| ACCENT | 0.13 | 0.28 | 0.16 | 0.32 | 0.18 | 0.38 | 0.87 | 0.62 | 0.21 |
| $LXR_{pos}$ | 0.04 | 0.53 | 0.058 | 0.603 | 0.075 | 0.66 | 0.971 | 0.43 | 0.14 |
| $LXR_{neg}$ | 0.031 | 0.70 | **0.046** | **0.781** | **0.056** | 0.82 | 0.987 | 0.232 | 0.13 |
| **LXR** | **0.03** | **0.71** | 0.048 | 0.78 | 0.057 | **0.83** | **0.99** | **0.23** | **0.12** |

**Table 2: AUC values for explaining a VAE recommender.**

| Method | k=5 | | k=10 | | k=20 | | INS | DEL | NDCG |
|---|---|---|---|---|---|---|---|---|---|
| | POS | NEG | POS | NEG | POS | NEG | | | |
| Jaccard | 0.32 | 0.81 | 0.06 | **0.88** | 0.52 | **0.91** | 0.015 | 0.003 | 0.35 |
| Cosine | 0.64 | 0.67 | 0.73 | 0.75 | 0.79 | 0.81 | 0.009 | 0.004 | 0.35 |
| LIME | 0.54 | 0.54 | 0.46 | 0.88 | 0.55 | 0.90 | 0.012 | 0.0057 | 0.47 |
| SHAP | 0.53 | 0.54 | 0.62 | 0.63 | 0.70 | 0.71 | 0.008 | 0.007 | 0.41 |
| ACCENT | 0.37 | 0.58 | 0.45 | 0.70 | 0.53 | 0.79 | 0.013 | 0.0045 | 0.41 |
| $LXR_{pos}$ | 0.29 | 0.82 | 0.40 | 0.86 | 0.50 | 0.88 | **0.019** | 0.004 | 0.34 |
| $LXR_{neg}$ | 0.55 | 0.56 | 0.66 | 0.67 | 0.75 | 0.76 | 0.009 | 0.0067 | 0.52 |
| **LXR** | **0.27** | **0.83** | **0.37** | 0.87 | **0.46** | 0.89 | 0.013 | **0.002** | **0.33** |

In conclusion, we focus on the outcomes of our ablation test, juxtaposing LXR with its two ablated versions, namely $LXR_{pos}$ and $LXR_{neg}$. It's evident that in the majority of scenarios (six out of the nine tests for each dataset), the complete LXR model outperforms its ablated counterparts. In Table 1, $LXR_{neg}$ seems to have a slight edge over $LXR_{pos}$, but the opposite appears to be the case in Table 2. It appears that the two objective terms, $\mathcal{L}_{pred}$ and $\mathcal{L}_{inv}$ from Eq. 1, are equally usefull. Nonetheless, these findings accentuate the significance of integrating both terms into LXR's objective, leading to superior overall outcomes.

### 5.4 Sanity Test for Explanation Methods

Adebayo et al. [3] showed that some explanation methods, while able to generate accurate explanations, are in fact insensitive to the model being explained. For instance, an explanatory model might justify a recommended item based on its external similarity to items from the user's past interactions. Although this might seem like a coherent explanation to the user, it doesn't originate from the recommendation model under scrutiny. A comparable observation has been made in multiple prominent explanation methods designed for computer vision models [3].

In order to address the above issue, the *parameter randomization* sanity test was proposed in order to evaluate the sensitivity of explanation models to the model being explained. In our case, the test entails comparing the explanations generated by LXR in two distinct configurations: (1) a "normal" setup where the explained model is an optimized VAE recommendation model on the ML1M dataset, and (2) an "altered" setup, in which the same recommender is used, but with the weights of its last $K$ layers scrambled. Figure 4 depicts the results of this test. To compare explanations from both

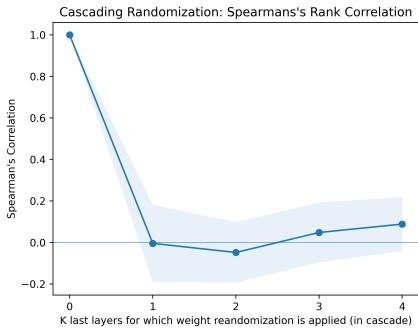

**Figure 4: The parameter randomization sanity test for explanation models [3]. At $K = 0$ no randomization is employed. As the randomization process begins, the correlation immediately drops indicating the high sensitivity of LXR to the model being explained.**

cases, we employ Spearman's correlation on the ranked list of user data (items) according to the explainability score given by LXR in each setting. At $K = 0$, no randomization is employed, hence the correlation is maximal. However, as the randomization process begins, the correlation swiftly drops to zero. Such a result underscores LXR's profound sensitivity to the model being explained.

## 6 CONCLUSION

The LXR framework, outlined in this paper, is a post-hoc, model-agnostic approach for counterfactual explanations of recommender systems. Anchored around a novel counterfactual objective, LXR's self-directed learning seeks to pinpoint the most important user data with respect to a recommended item. To achieve this, LXR adopts a non-conventional design where the recommender system is intrinsically intertwined within the learning objective and plays a proactive role during optimization. A key advantage of LXR when compared to alternative counterfactual explanation techniques, lies in its efficiency to rapidly generate explanations, bypassing the necessity for intensive computations or perturbations. This proficiency positions LXR as a premier choice for delivering on-demand explanations in real-world systems. Lastly, we introduce a set of novel evaluation metrics specifically designed for counterfactual evaluation of recommender systems. These metrics serve as an additional contribution of this study, providing a valuable resource for future researchers to assess and evaluate explanation methods for recommender systems.

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

## 7 APPENDIX

In the appended section, we provide supplementary evaluation outcomes using the following datasets:

**Yahoo! Music (Yahoo!):** A random sample of 13,725 users with 10,265 items and 687,590 ratings from the Yahoo! Music dataset [15].

**Pinterest:** A random sample of 19,155 users with 9,639 items and 500,000 ratings from Pinterest dataset [21].

Tables 3 and 4 present the AUC results for explaining an MF and a VAE recommender on the Yahoo! dataset, respectively. Tables 5 and 6 present the AUC results for explaining an MF and a VAE recommenders on the Pinterest dataset, respectively. The tables are similar in structure to Tabs. 1-2 from Sec. 5. They depict the AUC values for *POS-P@K* and *NEG-P@K* with $K = 5, 10, 20$. Additionally, values for the AUC of *INS-P*, *DEL-P*, and *NDCG-P* are provided for LXR and the baselines. On each evaluation, the best result is emphasized with **bold** fonts, while the second-best result is underlined. The distinction in results between the top model and the next best was statistically verified using a t-test, yielding a p-value of less than 0.05. These results strengthen the reliability and further emphasize the advantage of the LXR framework with respect to its alternatives.

**Table 3: MF recommender on the Yahoo! dataset.**

| Method | k=5 | | k=10 | | k=20 | | INS | DEL | NDCG |
|---|---|---|---|---|---|---|---|---|---|
| | POS | NEG | POS | NEG | POS | NEG | | | |
| Jaccard | 0.19 | 0.36 | 0.27 | 0.44 | 0.34 | 0.52 | 0.97 | 0.85 | 0.28 |
| Cosine | 0.27 | 0.28 | 0.36 | 0.37 | 0.43 | 0.44 | 0.95 | 0.88 | 0.34 |
| LIME | *0.14* | 0.49 | *0.18* | 0.56 | *0.23* | 0.62 | 0.98 | 0.71 | 0.18 |
| SHAP | 0.28 | 0.27 | 0.32 | 0.31 | 0.38 | 0.37 | 0.94 | 0.83 | 0.33 |
| ACCENT | 0.18 | 0.39 | 0.22 | 0.44 | 0.27 | 0.49 | 0.97 | 0.73 | 0.26 |
| LXR_pos | 0.099 | 0.57 | 0.13 | 0.63 | 0.17 | 0.68 | 0.989 | 0.63 | 0.19 |
| LXR_neg | 0.06 | 0.69 | 0.08 | 0.74 | 0.11 | 0.78 | 0.998 | 0.41 | 0.14 |
| **LXR** | **0.058** | **0.70** | **0.07** | **0.75** | **0.10** | **0.79** | **0.99** | **0.40** | **0.13** |

**Table 4: VAE recommender on the Yahoo! dataset.**

| Method | k=5 | | k=10 | | k=20 | | INS | DEL | NDCG |
|---|---|---|---|---|---|---|---|---|---|
| | POS | NEG | POS | NEG | POS | NEG | | | |
| Jaccard | 0.33 | 0.74 | 0.42 | 0.81 | 0.52 | 0.86 | 0.03 | 0.009 | 0.36 |
| Cosine | 0.54 | 0.55 | 0.63 | 0.64 | 0.71 | 0.72 | 0.02 | 0.02 | 0.51 |
| LIME | 0.38 | 0.75 | 0.47 | **0.82** | 0.57 | 0.87 | 0.03 | 0.01 | 0.45 |
| SHAP | 0.53 | 0.47 | 0.62 | 0.56 | 0.71 | 0.65 | 0.02 | 0.013 | 0.49 |
| ACCENT | 0.48 | 0.52 | 0.56 | 0.66 | 0.65 | 0.76 | 0.024 | 0.013 | 0.48 |
| LXR_pos | 0.30 | **0.76** | 0.39 | 0.81 | 0.51 | 0.84 | **0.04** | 0.0088 | 0.345 |
| LXR_neg | 0.35 | 0.69 | 0.45 | 0.76 | 0.56 | 0.81 | 0.03 | 0.01 | 0.38 |
| **LXR** | **0.295** | 0.73 | **0.37** | 0.79 | **0.47** | **0.88** | 0.029 | **0.0087** | **0.342** |

**Table 5: MF recommender on the Pinterest dataset.**

| Method | k=5 | | k=10 | | k=20 | | INS | DEL | NDCG |
|---|---|---|---|---|---|---|---|---|---|
| | POS | NEG | POS | NEG | POS | NEG | | | |
| Jaccard | 0.25 | 0.24 | 0.27 | 0.26 | 0.29 | 0.29 | 0.96 | 0.89 | 0.33 |
| Cosine | 0.24 | 0.24 | 0.26 | 0.26 | 0.29 | 0.30 | 0.96 | 0.90 | 0.33 |
| LIME | 0.12 | 0.48 | 0.13 | 0.51 | 0.16 | 0.55 | 0.99 | 0.70 | 0.20 |
| SHAP | 0.27 | 0.26 | 0.29 | 0.28 | 0.32 | 0.31 | 0.95 | 0.90 | 0.34 |
| ACCENT | 0.27 | 0.26 | 0.29 | 0.29 | 0.32 | 0.29 | 0.93 | 0.88 | 0.34 |
| LXR_pos | 0.11 | 0.46 | 0.12 | 0.48 | 0.15 | 0.52 | 0.997 | 0.75 | 0.19 |
| LXR_neg | 0.17 | 0.42 | 0.18 | 0.45 | 0.21 | 0.49 | 0.98 | 0.68 | 0.25 |
| **LXR** | **0.10** | **0.56** | **0.11** | **0.59** | **0.14** | **0.64** | **0.998** | **0.53** | **0.18** |

**Table 6: VAE recommender on the Pinterest dataset.**

| Method | k=5 | | k=10 | | k=20 | | INS | DEL | NDCG |
|---|---|---|---|---|---|---|---|---|---|
| | POS | NEG | POS | NEG | POS | NEG | | | |
| Jaccard | 0.28 | 0.70 | 0.35 | 0.69 | 0.51 | 0.81 | 0.018 | 0.007 | 0.44 |
| Cosine | 0.46 | 0.57 | 0.55 | 0.67 | 0.64 | 0.74 | 0.013 | 0.008 | 0.45 |
| LIME | 0.32 | 0.70 | 0.41 | 0.76 | 0.50 | 0.816 | 0.017 | 0.006 | 0.35 |
| SHAP | 0.49 | 0.48 | 0.58 | 0.57 | 0.67 | 0.65 | 0.012 | 0.009 | 0.40 |
| ACCENT | 0.36 | 0.46 | 0.44 | 0.59 | 0.51 | 0.69 | 0.015 | 0.007 | 0.39 |
| LXR_pos | **0.26** | 0.71 | 0.35 | 0.77 | 0.45 | 0.82 | **0.02** | **0.0053** | 0.39 |
| LXR_neg | 0.27 | 0.71 | 0.345 | 0.76 | 0.46 | 0.84 | 0.19 | 0.0054 | 0.35 |
| **LXR** | 0.30 | **0.81** | **0.34** | **0.8** | **0.42** | **0.85** | 0.019 | 0.0059 | **0.34** |

