# OpenReview forum: "Learning Counterfactual Explanations for Recommender Systems"
_ACM.org/TheWebConf/2024/Conference — TheWebConf24_

### Official Review · Reviewer_daGC · 2023-11-22

**Novelty:** 3
**Technical Quality:** 4

**Review:**

This manuscript proposes a generic architecture for explain-Learning to eXplain Recommendation (LXR), LXR can work with any differentiable recommender and learns to rate the importance of the user's data relative to the recommended items, LXR is proposed to identify the most important user data related to the recommended items. Also, the authors designed four sets of metrics for performance evaluation, providing a referable method for future researchers to evaluate the interpretation methods of recommender systems. In comparison with other methods, LXR has a better performance.

Strengths:
The recommendation interpretation framework proposed by the authors is generalizable and can be used after any microable recommendation network. The four sets of metrics proposed are interesting and fully take into account the explanatory aspects of the evaluation.

Weakness:
Some expressions in the text need to be explained. Most of the references cited are before 2020, which is a bit older. The baseline for comparison please preferably choose the most recent one for comparison.

**Questions:**

1. Why does "y" in Figure 1 only point to pred and not to inv and mask, and why is "" used in the formula and "L" in Figure 1?
2. Equation 1 in section 3.2, please explain how the two hyperparameters 𝜆_𝑖𝑛v and 𝜆_mask are chosen, adaptive or fixed parameters?
3. section 4 before 4.1 can be considered in related work.
4. In the introduction about the indicator POS-P@K, in the formula for AUC POS-P@K, it should be POS-P@K (m), you are missing a "-P".
5. Section 5.4 needs more clarification, it's a bit too succinct.
6. When I run cell 11 of “LXR_MLP_ML1M.ipynb”, it prompts for no “shap_values.pkl” file, and when I run cell 3 of “LXR_VAE_metrics.ipynb”, it prompts for no “train_data.csv” file, please provide detailed instructions on how to run in 'README'. Why not use the ".py" program instead of jupyter notebook?

**Reviewer Confidence:**

3: The reviewer is confident but not certain that the evaluation is correct

**Scope:**

3: The work is somewhat relevant to the Web and to the track, and is of narrow interest to a sub-community

---

### Official Review · Reviewer_9RBP · 2023-11-22

**Novelty:** 5
**Technical Quality:** 5

**Review:**

This paper presents Learning to eXplain Recommendations (LXR) for counterfactual explanations of recommender systems. It is designed to be post-hoc and model-agnostic (treating the recommender systems as a black box which when given user interaction history as input outputs recommendation). The basic idea behind it is to learn a explainer model which takes interaction history and a target item (that's being recommended) and outputs a masking mechanism indicating the important items in the interaction history to explain the recommended target item. The objective for the explainer model consists of two parts + l1 regularization, one part encourages the masking to help with recommending the target item and the other part penalizes the inverse masking (1-mask) for recommending the target item. Experimental results demonstrate that LXR does what it is set out to do.

Overall the paper is clearly written and easy to follow. There are some minor notational inconsistency/confusion which I will list below, but nothing major. The idea behind LXR is intuitive and seems novel, though I am not very familiar with explainable recommendation literature therefore I am afraid my assessment on this would be fairly low confidence. The experimental results seem convincing (though i have more thoughts on that later). I have two relatively high-level (meta)-questions:

- I can imagine evaluation being very difficult for explainable recommendation. I am certainly no experts on this and I am happy to listen to other reviewers' thoughts on how valid the evaluation is done in this paper. However, from my understanding, isn't the model objective specifically designed to make the metrics used in this paper look good? A common theme of all the metrics used in this paper is that they try to make the recommender stay stable when deleting "unimportant" items and change rapidly when deleting "important" items. This is exactly what the two parts of LXR objective are designed to address. Therefore, I am not surprised that LXR outperforms other baselines on these metrics. However, are there any other types of metrics used in the explainable literature which doesn't focus on this aspect but is also widely acknowledged? How would LXR perform on these metrics?

- If my understanding is correct, from line 352 left column, the mask vector is always element-wise multiplied with interaction history x, meaning it can never infer explanation from the unlabeled part of the data. Given how important the unlabeled part of the data is for implicit feedback CF, this feels like a missed opportunity. Are there any specific reasons for this choice? On a related note, isn't "counterfactual" implying that we should look at both turning some of the 1's to 0's and turning some of the 0's to 1's equally?

Minor:
- Line 195 right column: the function should be mapped to a simplex as opposed to [0, 1]^V, if the output represents "a probability distribution over the items" (which is what VAE is doing).

- The term "mask" is often used to refer to binary mask (at least in the neural network context) hence it might be better to clarify that when introducing the vector m.

- Eq(1) writes the second term as L_{inv} (f(x^{1-m}, y) but later L_{inv} is defined as log(1 - f(x^{1-m}) -- 1-m appearing in both will cancel each other which I suppose is not intended.

**Questions:**

(Copied from above)

- I can imagine evaluation being very difficult for explainable recommendation. I am certainly no experts on this and I am happy to listen to other reviewers' thoughts on how valid the evaluation is done in this paper. However, from my understanding, isn't the model objective specifically designed to make the metrics used in this paper look good? A common theme of all the metrics used in this paper is that they try to make the recommender stay stable when deleting "unimportant" items and change rapidly when deleting "important" items. This is exactly what the two parts of LXR objective are designed to address. Therefore, I am not surprised that LXR outperforms other baselines on these metrics. However, are there any other types of metrics used in the explainable literature which doesn't focus on this aspect but is also widely acknowledged? How would LXR perform on these metrics?

- If my understanding is correct, from line 352 left column, the mask vector is always element-wise multiplied with interaction history x, meaning it can never infer explanation from the unlabeled part of the data. Given how important the unlabeled part of the data is for implicit feedback CF, this feels like a missed opportunity. Are there any specific reasons for this choice? On a related note, isn't "counterfactual" implying that we should look at both turning some of the 1's to 0's and turning some of the 0's to 1's equally?

**Reviewer Confidence:**

3: The reviewer is confident but not certain that the evaluation is correct

**Scope:**

3: The work is somewhat relevant to the Web and to the track, and is of narrow interest to a sub-community

---

### Official Review · Reviewer_8wcz · 2023-11-23

**Novelty:** 3
**Technical Quality:** 4

**Review:**

This paper studies the couterfactual explainalbe recommendation problem, which aims to idenfiy the key personal information of users for maintaining same recommendation results (if altered, the recommendation results would change). The main contributions lie in two aspects. Firstly, it proposes an efficient, post-hoc, model-agnostic framework (called LXR) for counterfactual explanations. It only requires the differentiability of the recommendation models. Secondly, it proposes several counterfactual evaluation metrics for assessing explanations in recommender systems.
Strengths
1. The proposed method is efficient and model-agnostic, which makes it suitable for delivering on-demand explanations in various recommender systems.
2. The proposed method is well-explained and easily to understand.
3. The source code is publicly available.
Improvements
1. The proposed method is not dependent on the target item, which makes it unreasonable.
2. The efficiency of LXR is not thoroughly explored both empirically and theoretically, though it is claimed that it is efficient for on-demand scenarios.
3. The motivation of such a method is not well described.

**Questions:**

1. The proposed method is not dependent on the target item, which makes it unreasonable. It means that the explanations of LXR for all items with the same label are the same, which actually contradicts the real-world scenarios. It also makes me doubt the motivation for such an algorithm.
2. The efficiency of LXR is not thoroughly explored both empirically and theoretically, though it is claimed that it is efficient for on-demand scenarios. There should be an analysis of the time complexity of the proposed method aganist other baselines, and a comparison of the experimental time of these methods.
3. The application scenarios of such counterfactual explanantions should be thorough described. I doubt the practicability of the formulated problem, even though I set aside the problem of LXR itself as pointed above.

**Reviewer Confidence:**

3: The reviewer is confident but not certain that the evaluation is correct

**Scope:**

4: The work is relevant to the Web and to the track, and is of broad interest to the community

---

### Official Review · Reviewer_xr8y · 2023-11-23

**Novelty:** 3
**Technical Quality:** 5

**Review:**

The paper introduces the LXR framework, a novel, model-agnostic approach for generating counterfactual explanations in recommender systems. The framework also introduces new evaluation metrics for assessing the quality of explanations in this context.

Strength:
- The proposed method can rapidly generate explanations.
- The authors propose several evaluation metrics for explanations.
- Open sources implementation code to support reproducibility.

Weakness:
- The novelty is relatively weak. Too much work on counterfactual explanations for recommendation.
- The idea is similar to combining factual and counterfactual explanations in GNN. Some related works are missing, for example:
	- Chen, Ziheng, et al. "Grease: Generate factual and counterfactual explanations for gnn-based recommendations."
	- Tan, Juntao, et al. "Learning and evaluating graph neural network explanations based on counterfactual and factual reasoning."
	- Wang, Danqing, et al. "Generating Global Factual and Counterfactual Explainer for Molecule under Domain Constraints."
- It is uncertain whether the evaluation method proposed in this paper can be applied to a broader range of counterfactual explanations, such as counterfactual on features, paths in graphs, etc.

**Questions:**

Please see weakness above.

**Reviewer Confidence:**

3: The reviewer is confident but not certain that the evaluation is correct

**Scope:**

3: The work is somewhat relevant to the Web and to the track, and is of narrow interest to a sub-community

---

### Official Review · Reviewer_dg9B · 2023-11-27

**Novelty:** 5
**Technical Quality:** 4

**Review:**

The paper introduces a new model-agnostic methodology, LXR, for creating counterfactual explanations for recommendations systems.  The authors claim a number of innovations – self-supervised counterfactual loss term, for example – that allow for a model-agnostic approach.  And one that does not require computer intensive calculations, which can be a potentially significant advantage.

This is potentially very interesting for generic capabilities for explanation.  The main question that was not broached in detail was, is a model-agnostic approach really required and fit for purpose? The authors show good results against baselines of Jaccard, Cosine similarity, SHAP, LIME and Accent.  This is a reasonable selection, but there was no comparison to a specific baseline that may be implemented in a commercial environment.  The use case mentioned in the introduction is for GDPR.  Is the convenience of a model-agnostic, low-computational approach overshadowed by regulatory requirements, where a best in class approach may be preferred?  We cannot answer from this paper.

Four new metrics were introduced to gauge the effectiveness of the authors’ methods.  But there were no human trials. Given that the goal is to show explainability, which inherently is a human interpretation, with only backtesting against standard but not exhaustive databases (MovieLens, Yahoo!Music and Pinterest) the paper falls short.  Especially since the metrics are new themselves and have no history associated with them.

**Questions:**

What is the possibility to test the LXR approach against in-place commercial ones, and to perform testing on people (versus backtesting data)?

**Ethics Review Description:**

No issue

**Reviewer Confidence:**

3: The reviewer is confident but not certain that the evaluation is correct

**Scope:**

4: The work is relevant to the Web and to the track, and is of broad interest to the community

---

### Decision · Program_Chairs · 2024-01-22

**Decision:**

Accept

**Comment:**

The paper presents a model-agnostic explanation method for recommender systems based on the idea of counterfactual explanation. This is realized by learning perturbations over the interaction history and the recommended item through a masking mechanism to find the important items in the interaction history. Overall the paper makes a reasonable contribution in terms of model-agnostic explainable recommendation. During discussions, authors made clarification in terms of the model design, efficiency, and evaluation. Authors are advised to add the clarifications, missing citations and possibly other evaluation methods such as sufficiency/necessity-based metrics into the paper.